# Uncovering Disease-Related Polymorphisms through Correlations between SNP Frequencies, Population and Epidemiological Data

**Samara Marques Dos Reis** [1,2], **Cristhian Augusto Bugs** [1], **José Artur Bogo Chies** [2,*]
and **Andrés Delgado Cañedo** [1]

1 Centro de Pesquisa em Biotecnologia, Universidade Federal do Pampa,
São Gabriel 96460, Rio Grande do Sul, Brazil

2 Laboratório de Imunobiologia e Imunogenética, Programa de Pós-Graduação em Genética e Biologia Molecular, Departamento de Genética, Universidade Federal do Rio Grande do Sul,
Porto Alegre 15053, Rio Grande do Sul, Brazil

* Correspondence: jabchies@terra.com.br; Tel.: +55-51-999-603-008

**Abstract:** Background: According to GWAS, which analyzes large amounts of DNA variants in case-control strategies, the genetic differences between two human individuals do not exceed 0.5%. As a consequence, finding biological significance in GWAS results is a challenging task. We propose an alternative method for identifying disease-causing variants based on the simultaneous evaluation of genome variant data acquired from public databases and pathology epidemiological data. This method is grounded on the following premise: If a particular pathology is common in a community, genetic variants that confer susceptibility to that pathology should also be common in that population. Methods: Three groups of genes were evaluated to test this premise: variants related to depression found through GWAS, six genes unrelated to depression, and four genes already genotyped in case-control studies involving depression (*TPH2*, *NR3C1*, *SLC6A2* and *SLC6A3*). In terms of GWAS depression-related variants, nine of the 82 SNPs evaluated showed a favorable correlation between allele frequency and epidemiological data. As anticipated, none of the 286 SNPs were correlated in the neutral group. In terms of proof of concept, two THP2 variants, 26 *NR3C1* variants and four *SLC6A3* variants were found to be related to depression rates and epidemiological statistics. Conclusions: Together with data from the literature involving these SNPs, these correlations support this strategy as a complementary method for identifying possible disease-causing variants.

**Keywords:** depression; variants; polymorphisms; genome projects; biodate; databases

## 1. Introduction

Several molecular approaches to detection and analysis of DNA variants that have been developed in recent years have altered the dynamics of genetic studies, enabling the identification of slight differences both within and between human population groups. In medical approaches, the study of genetic variation comparing cases and controls allows for identifying specific genetic variants associated with diseases, which can be used as biomarkers for early diagnosis or to monitor progression of the disease [1]. A vast amount of information regarding human DNA variants is currently stored in genetic databases accessible to the scientific community. The 1000 Genomes Project stands out among these databases because it provides access to information about human genetic variants, such as single nucleotide polymorphisms (SNPs) and structural variations. This database also allows for an evaluation of genetic variant linkage disequilibrium. High-coverage (30X) allelic and genotypic frequencies were available for 2054 individuals from 26 human populations around the world [2–5] until August 2020, when 698 additional samples were added [6]. The 1000 Genomes Project was designed to create public genome-wide databases

and is frequently used in conjunction with genome-wide association studies (GWAS), a broader method that identifies SNPs associated with disease susceptibility or resistance by genotyping hundreds of thousands of loci using a case–control strategy [7,8].

Currently, a number of monogenic diseases are readily diagnosed, allowing for prompt and precise treatment recommendations. Nonetheless, a significant number of diseases have a complex genetic component, indicating a multifactorial etiology [9].

Depression is one of the complex diseases for which GWAS approaches have already provided some information about potential variants and loci associated with the disease, and more consistent results are obtained as the number of patients tested grows. Despite this, the identification of more than 80 loci associated with depression provides a number of challenges to the study of depression-associated variants [10,11]. Among them are the need to prioritize causal genes and the significance of allelic frequency, primarily for establishing the genetic architecture of depression. In addition, GWAS cannot explain specific biological characteristics of the disease, such as the physiological mechanisms beyond a given association; thus, additional approaches are necessary to meet such demands.

Taking into account the vast amount of information on human DNA variants and global epidemiological data available in various databases and databanks pertaining to a variety of pathologies, we propose the following evaluation: If a given pathology is highly prevalent in a particular population, then genetic variants conferring susceptibility to this disease should also be prevalent in that population.

In the current study, our hypothesis is examined utilizing data on major depression (OMIM 608516) from the World Health Organization (WHO) and variant databases such as the 1000 Genomes Project. As already mentioned, genetic variants associated with depression in the genetic variant database should be overrepresented in human populations with high depression rates, whereas genetic variants not associated with depression should not show differences between these population groups. In contrast, populations with low depression rates should also have low frequencies of genetic variants already associated with the disease. As a proof of concept, we chose four genes (*TPH2*, *NR3C1*, *SLC6A2* and *SLCA3*) that had previously been genotyped in case–control studies involving depression. If our hypothesis is correct (i.e., if genetic variants associated with depression are overrepresented in human populations with high depression rates), then this new method would provide evidence for this claim.

## 2. Materials and Methods

### 2.1. Choosing Allelic Variants and Obtaining Their Frequencies for Different Human Populations

Three groups of genes were analyzed to assess the correlation hypothesis: (a) variants associated with depression identified in genome-wide association studies (GWAS), from Howard, Amare, Wray and Hyde [11–14]; (b) variants in genes not previously associated with depression (*ACT*, *B2M*, *EEF2*, *GAPDH*, *PLEK2* and *PPIA*); and (c) variants in four genes already examined in case–control studies involving depression (*TPH2*, *NR3C1*, *SLC6A2* and *SLCA3*). Using the Variant Annotation package in R software to process the VCF files [15], allele and genotype frequencies of these variants were obtained for different populations available through the 1000 Genomes Project (https://www.internationalgenome.org/) accessed on 5 May 2021.

### 2.2. Statistical Analysis

Using R software, the allelic and genotype frequencies of each population were subjected to Pearson's correlation test against the depression rates inferred for each population. The depression rates for each human population were obtained from the World Health Organization [16].

The analysis was performed with a 99% confidence level ($p \leq 0.01$). Due to linkage disequilibrium (LD), the SNPs tested were not entirely independent; therefore, $p$-value correction was not applied. In this way, the potential inherent uncertainties resulting from multiple tests were taken into account when interpreting the results.

The maximum allele frequency (MAF) was calculated using three models: first, the maximum allele frequency was exaggerated when a monogenic recessive model with complete penetrance was considered, where MAF = (depression rate). We also considered the Whiffin et al. [17] model, which considers MAF < 0.10, and the Sullivan et al. [18] model, which considers MAF < 0.01. The hypothesis testing is described in the flowchart presented in Figure 1.

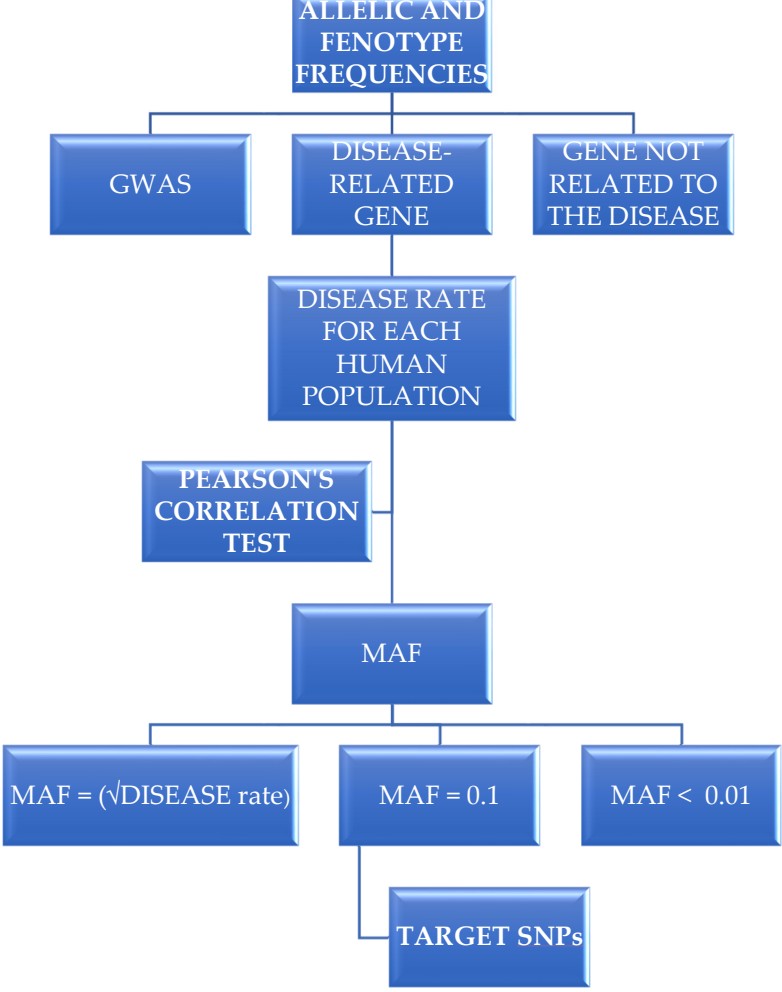

**Figure 1.** Flowchart representing the hypothesis testing. MAF = maximum allele frequency, used to filter SNPs after Pearson's correlation test.

### 2.3. Linkage Disequilibrium

The genetic SNPs that showed a correlation with depression were evaluated using the LDlink (https://ldlink.nci.nih.gov/), accessed on 20 July 2021, online suite tools [19]. To evaluate SNPs with complete linkage, they were submitted to the LDHap tool with all populations selected for analysis. The SNPs that did not demonstrate complete linkage were evaluated in pairs using LDpair with the All Populations option selected. The association of these SNPs with GWAS-studied traits was then evaluated using LDassoc [20], and the association with changes in gene expression was assessed using LDexpress, selecting all tissues in all populations and setting an R2 value > 0.3, $p < 0.1$, and linkage with SNPs located up to 50,000 bases. SNPclip was then used to evaluate the presence of SNPs potentially associated with depression in commercial genotyping platforms.

## 3. Results

### 3.1. The Correlation Approach Detected Significant Variants in GWAS-Associated Data But Not in the Postulated Unrelated Genes

Using the correlation method, a total of 82 SNPs in genes potentially associated with depression were tested against depression rates. Nine of these eighty-two SNPs (10.97%) correlate positively with the depression rates in the different populations. After stratification according to different maximum allele frequency (MAF) models, one SNP presented a positive correlation with depression, with an allele frequency between 0.01 and 0.10 (rs17727765), and another eight SNPs showed correlations with depression, with allele frequencies ranging from 0.10 to 0.33. Table 1 provides specific details about the SNPs that correlate. Table S1 presents the correlation values for all variants associated with depression in GWAS.

**Table 1.** Distribution of allele frequencies corresponding to the GWAS variants correlated with depression.

| Alternative Allele Frequency | Total | sigRR | sigAA | SNP (Allele Frequency/Correlation *p*-Value) * |
|:---:|:---:|:---:|:---:|:---:|
| <0.01 | 0 | 0 | 0 | ------- |
| 0.01–0.1 | 4 | 0 | 1 | rs17727765 (AFr = 0.03, *p* = 0.004); |
| 0.1–0.33 | 22 | 0 | 4 | rs7200826 (AFr = 0.15, *p* = 0.003); rs112348907 (AFr = 0.27, *p* = 0.007); rs2422320(AFr = 0.29, *p* = 0.0002); rs2422321 (AFr = 0.29, *p* = 0.0001) |
| 0.33–0.66 | 38 | 0 | 0 | ------- |
| 0.66–0.9 | 18 | 4 | 0 | rs10929355 (AFr = 0.69, *p* = 0.010); rs4904738 (AFr = 0.71, *p* = 0.002); rs7044150 (AFr = 0.84, *p* = 0.003); rs1354115 (AFr = 0.81, *p* = 0.004) |
| 0.9–0.99 | 0 | 0 | 0 | ------- |
| >0.99 | 0 | 0 | 0 | ------- |

* SNPs filtered by the frequency of the allele associated with depression rates. AFr = Allele frequency of the correlated allele.

To evaluate variants of genes not previously associated with depression, we chose genes frequently used as normalizers in studies of relative gene expression: *ACTB* (21 SNPs), *B2M* (36 SNPs), *EEF2* (68 SNPs), *GAPDH* (39 SNPs), *PLEK2* (121 SNPs) and *PPIA* (121 SNPs). Using the allele and genotype frequencies of these 331 variants, the correlation test reveals that none of them are associated with depression rates. Table S2 displays the results of the correlation test using variants of these genes.

### 3.2. Proof of Concept: The TPH2, NR3C1 and SLC6A3, but Not the SLC6A2 Gene Variants, Were Associated with Depression

The correlation method was applied to variants of four genes previously investigated in case–control studies in relation to depression. The correlation between 550 variants of the *TPH2* gene and depression rates in human populations from around the world was examined. According to the results, shown in Table S3, two SNPs showed a positive correlation (0.4%). As shown in Table 2, one SNP (rs7298203) could be correlated with depression if MAF < 0.1, and one SNP (rs4760820) could be potentially correlated if MAF < 0.3 is considered.

The evaluation of 562 variants of the *NR3C1* gene is detailed in Table S4. As shown in Table 3, 26 SNPs (4.6%) correlated with depression, with 11 SNPs presenting MAF 0.1 and 15 SNPs potentially correlating using the most permissive model after maximum allele frequency filtering, with MAF 0.3.

Four *SLC6A3* gene SNPs (1.1%) were associated with depression (rs13189021, rs10052016, rs62331084 and rs10053602); in all cases, the frequencies of the alleles involved were within the most permissive model (MAF < 0.33), as shown in Table 4. The information regarding the *SLC6A2* and *SLC6A3* gene variants is shown in Tables S5 and S6, respectively.

**Table 2.** Distribution of the allele frequency corresponding to the *TPH2* gene variants correlated with depression.

| Alternative Allele Frequency | Total | sigRR | sigAA | SNP (Allele Frequency/Correlation *p*-Value) * |
|---|---|---|---|---|
| <0.01 | 103 | 0 | 0 | ------- |
| 0.01–0.1 | 162 | 0 | 0 | ------- |
| 0.1–0.33 | 101 | 0 | 1 | rs4760820 (AFr = 0.179. *p* = 0.002); |
| 0.33–0.66 | 122 | 0 | 5 | ------- |
| 0.66–0.9 | 57 | 0 | 0 | ------- |
| 0.9–0.99 | 3 | 1 | 0 | rs7298203 (AFr = 0.974. *p* = 0.008) |
| >0.99 | 0 | 0 | 0 | ------- |

* SNPs filtered by the allele frequencies of the allele correlated with depression rates, according to the models reported in the "Materials and methods" section.

**Table 3.** Distribution of the allele frequency corresponding to the *NR3C1* gene variants correlated with depression.

| Alternative Allele Frequency | Total | sigRR | sigAA | SNP (Allele Frequency/Correlation *p*-Value) * |
|---|---|---|---|---|
| <0.01 | 225 | 2 | 3 | ------- |
| 0.01–0.1 | 167 | 0 | 11 | rs116798177 (AFr = 0.02, *p* = 0.004); rs142327762 (AFr = 0.04, *p* = 0.002); rrs61752263 (AFr = 0.04, *p* = 0.002); rs55817235 (AFr = 0.04, *p* = 0.002); rrs56150733 (AFr = 0.04, *p* = 0.002); rs72801051 (AFr = 0.05, *p* = 0.002); rrs72801054 (AFr = 0.05, *p* = 0.002); rs141755899 (AFr = 0.05, *p* = 0.002); rrs72801080 (AFr = 0.05, *p* = 0.002); rs10515522 (AFr = 0.05, *p* = 0.002); rrs72802806 (AFr = 0.09, *p* = 0.010) |
| 0.1–0.33 | 126 | 0 | 14 | rs258814 (AFr = 0.21. *p* = 0.004); rs13155635 (AFr = 0.28. *p* = 0.010); rs860457 (AFr = 0.21. *p* = 0.006); rs852979 (AFr = 0.21. *p* = 0.007); rs852982 (AFr = 0.21. *p* = 0.004); rs190488 (AFr = 0.21. *p* = 0.007); rs33380 (AFr = 0.21. *p* = 0.01); rs34158792 (AFr = 0.23. *p* = 0.020); rs61752282 (AFr = 0.21. *p* = 0.01); rs111440401 (AFr = 0.15. *p* = 0.008); rs1866388 (AFr = 0.21. *p* = 0.009); rs10053679 (AFr = 0.22. *p* = 0.012); rs41423247 (AFr = 0.25. *p* = 0.001); rs11747997 (AFr = 0.21. *p* = 0.009); |
| 0.33–0.66 | 27 | 0 | 0 | ------- |
| 0.66–0.9 | 6 | 1 | 0 | rs1837262 (AFr = 0.77. *p* = 0.01) |
| 0.9–0.99 | 11 | 0 | 0 | ------- |
| >0.99 | 0 | 0 | 0 | ------- |

* SNPs filtered by the allele frequencies of the allele correlated with depression rates, according to the models reported in the "Materials and methods" section.

**Table 4.** Distribution of the allele frequency corresponding to the *SLC6A3* gene variants correlated with depression.

| Alternative Allele Frequency | Total | sigRR | sigAA | SNP (Allele Frequency/Correlation *p*-Value) * |
|---|---|---|---|---|
| <0.01 | 40 | 0 | 0 | ------- |
| 0.01–0.1 | 183 | 0 | 0 | ------- |
| 0.1–0.33 | 80 | 0 | 4 | rs13189021 (AFr = 0.08, *p* = 0.001); rs10052016 (AFr = 0.18, *p* = 0.002); rs62331084 (AFr = 0.09, *p* = 0.003); rs10053602 (AFr = 0.18, *p* = 0.007); |
| 0.33–0.66 | 66 | 0 | 0 | ------- |
| 0.66–0.9 | 5 | 0 | 0 | ------- |
| 0.9–0.99 | 2 | 0 | 0 | ------- |
| >0.99 | 1 | 0 | 0 | ------- |

* SNPs filtered by the allele frequencies of the allele correlated with depression rates, according to the models reported in the "Materials and Methods" section.

*3.3. Linkage Disequilibrium and Variant Effects on Traits and Gene Expression*

According to the analysis, the correlation between the *TPH2* SNPs (rs7298203 and rs4760820) and depression rates (rs7298203 and rs4760820) was not strong ($R^2$: 0.0052). According to the LDtrait Tool, none of the variants discovered to be linked to diseases in GWAS studies are associated with disease ($R^2$: 0.0052). Although the rs4760820 variant is present on 25 of the 55 platforms analyzed with the SNPchip tool, the rs7298203 variant is absent from every platform.

When the variants of the *NR3C1* gene correlated with depression were evaluated with the LDhap tool, two blocks with complete linkage (D' = 1.0 and $R^2$ = 1.0), herein called Block 1 (rs10053679, rs34158792, rs860457, rs852979, rs258814, rs852982, rs190488, rs33380, rs61752282, rs1866388, rs11747997, rs1837262) and Block 2 (rs142327762, rs61752263, rs55817235, rs56150733, rs72801051, rs72801054, rs141755899, rs72801080, rs10515522) were revealed, as shown in Table 5. These blocks have a high D (D = 1.0) but a low $R^2$ (0.1642) value between them. Block 2 alleles with a lower frequency (0.044) are completely linked to Block 1 alleles with a lower frequency (0.221).

**Table 5.** Linkage disequilibrium of the variants correlated with depression.

| Gene | | Linkage SNPs |
|---|---|---|
| *TPH2* | | ----- |
| *NR3C1* | Block 1 | rs10053679, rs34158792, rs860457, rs852979, rs258814, rs852982, rs190488, rs33380, rs61752282, rs1866388, rs11747997, rs1837262 |
| | Block 2 | rs142327762, rs61752263, rs55817235, rs56150733, rs72801051, rs72801054, rs141755899, rs72801080, rs10515522 |
| *SLC6A3* | | rs10052016; rs10053602 |

Among the five variants outside these two blocks, rs111440401 displays high linkage disequilibrium with Block 1 variants (D': 0.9984 and $R^2$: 0.6589), and rs72802806 displays linkage with Block 2 variants (D' = 0.843 and $R^2$ = 0.356). Other variants (rs116798177, rs72802806, rs13155635, rs111440401 and rs41423247) were neither highly linked to variants of these two blocks nor to variants of the other block.

To analyze the association between *NR3C1* gene variants and variations in gene expression, the LDexpress tool was applied to one variant from each block and five variants outside the blocks. rs111440401 and the Block 1 variant were associated with alterations in *NR3C1* gene expression in the arteries and aorta. Concerning the rs41423247, rs13155635 and rs72802806 variants, the alleles associated with depression are associated with changes in the *NR3C1* expression level in the esophagus and mucosa. The Block 2 variants were unrelated to *NR3C1* expression changes.

The LDtrait evaluation revealed that the variants in Block 1 were associated with variations in hematocrit or hemoglobin concentrations, whereas no association was found for the Block 2 variants.

The rs111440401 variant, which was absent from all platforms evaluated with the SNPchip tool, could not be directly associated with trait or gene expression using the LDLink suite tools. The rs72802806 variant was associated with globulin-binding sex hormone levels, but it could only be evaluated on eight of the 55 platforms analyzed with SNPChip.

The analysis of *SLC6A3* variants revealed that rs10052016 and rs10053602 are completely linked, as shown in Table 5. The linkage disequilibrium between rs62331084 and rs13189021 is low, as with rs10052016/rs1003602. In addition, neither rs62331084 nor rs13189021 was associated with altered gene expression levels. On the other hand, rs10052016/rs10053602 inhibit the testicular expression of *SLC6A3*.

None of *SLC6A3* variants was directly involved with traits in the GWAS catalog according to the LDtrait tool, and the SNPchip tool showed that each of the four variants can only be evaluated in a few of the 59 platforms consulted; at the same time, the platforms that allow the analysis of rs10052016/rs10053602 (six and three platforms, respectively)

do not allow the analysis of rs13189021 (seven platforms) and rs62331084 (four platforms), and vice versa.

## 4. Discussion

Development of the Human Genome Project allowed for innovations in medical research, resulting in improved approaches to diagnosis, better understanding of the genetic basis of disease, and the creation of more effective treatments [21]. In this new medical context, understanding the genetic variation of individuals should be regarded as a crucial tool for both easing the quest for accurate and early disease diagnosis and driving the development of novel disease treatments.

Although the diagnosis and treatment of monogenic diseases are comparatively straightforward, this is not the case for diseases with multifactorial etiologies and complex genetic traits [9]. Currently, genome-wide association studies (GWAS) are the most prevalent technique for assessing such complex diseases. However, such research is quite costly and time-consuming, and its results cannot be extrapolated to other human populations. The majority of GWAS studies are conducted using individuals from a specific population or populations with a genetically related background. Consequently, these studies are frequently affected by the inherent bias of population stratification [22]. Therefore, a large sample size is required to obtain statistically significant results. Using 135,458 cases and 344,901 controls, a genome-wide association meta-analysis identified 44 independent and significant loci related to major depressive disorder [12]. In this regard, such methods are both costly and time-consuming.

GWAS is also used to identify correlations between genetic variants and traits, but it does not always provide obvious biological explanations, because a direct link between a genetic variant and a specific trait is not always known. This makes it more difficult to establish a direct connection between a gene and a particular mechanism or characteristic [23]. Therefore, complementary approaches to GWAS studies are welcome, such as using maximum allele frequency [17] and gene ontology or pathway-based tests [24].

In this research, we evaluated the logic underlying a complementary method for identifying DNA variants that may be associated with a particular phenotype. The proposal is predicated on the premise that if a given pathology is highly prevalent in a specific population, then genetic variants conferring susceptibility to this disease should also be prevalent in that population. Consequently, we hypothesize that the correlation of epidemiological data from various populations with their allelic and/or genotypic frequencies could reveal potential loci or variants that should be further studied as genetic risk factors for the disease in question. Consequently, we could concentrate on variants with high penetrance among the set of potentially involved variants.

Depression was chosen to evaluate our hypothesis because it is one of the most prevalent psychiatric disorders and stands out among the complex diseases with a significant impact on the global population. This disease has a multifactorial etiology, as both genetic and environmental factors contribute to its development [18]. Additionally, epidemiological data on depression are available for various human populations. Regarding genetic factors, four alternative genetic architecture models for depression have been proposed, namely: the rare allele model, the broad-sense heritability model, and the omnigenic model are concerned with allele frequencies, whereas the infinitesimal model is not (discussed by Ormel et al. [10]).

The environment plays a significant role in the development of depression, but genetics also plays a major part; therefore, our hypothesis predicts that the frequency of alleles previously associated with depression should be correlated with regional variations in the prevalence of depression. In light of the availability of epidemiological data regarding the prevalence of depression in various human populations around the world [16], we conducted a Pearson's correlation test against genetic variants clustered into three groups, as already mentioned. The first group consisted of more than eighty variants previously described as being associated with depression in GWAS [11–14], with the assumption

that they should have a positive correlation with global depression rates. The second group consisted of frequently used endogenous genes in gene expression experiments (ACT, B2M, EEF2, GAPDH, PLEK2 and PPIA) that were unrelated to depression. The assumption was that there would be no correlation between depression rates and this group, and, consequently, they would serve as a neutral or "negative control". As a final step, variants from four genes commonly evaluated in genotyping case–control studies involving individuals with major depression (*TPH2*, *NR3C1*, *SLC6A2* and *SLC6A3* genes) were analyzed as a confirmatory positive group, a proof-of-concept approach.

Before performing the analyses, three maximum allele frequency intervals were determined, taking into consideration the fact that this model targets genes with a significant effect on the heritability of depression. In these intervals, we used a maximum allele frequency of 0.33, which was derived from the global depression frequency, the limit of 0.1 based on data from Sullivan et al. [18], and 0.01 based on data from Whiffin et al. [17]. Due to the sample size of the 1000 Genomes Project, it would be impossible to use lower allelic frequencies in our analysis because the robustness of the analyses would be compromised. As additional populations with the same coverage level are added to this or similar databases in the future, it will be possible to further reduce these limits.

None of the 331 variants in the negative control group were associated with the epidemiological data. Notably, the results of this evaluation were used to research case–control studies in the scientific literature. However, none of the nine SNPs that our study found to be significant in case–control studies on depression had ever been evaluated.

Positively associated candidate genes from GWAS depression research revealed that nine of the 81 assessed variations were connected with depression rates; however, none of the 331 evaluated variants in the negative control group were associated with the epidemiological data. Notably, the findings of this evaluation were used to search the literature for information on case–control studies. However, none of the nine SNPs identified as relevant in our analysis had ever been examined in case–control studies of depression.

As proof of concept, the same method was used to evaluate different genetic variants of four commonly studied genes in case–control studies of depression. Two SNPs from the *TPH2* locus (rs4760820 and rs7298203) were found to be associated with depression rates; however, only rs4760820 presents literature-based evidence of an association. While Utge et al. [25] conducted a study in the Finnish population showing an association, Pereira et al. [26] found no significant association between rs4760820 and depression in a Brazilian cohort. Notably, the GG genotype was associated with other psychiatric conditions, such as bipolar disorder [27,28].

Concerning the *SLC6A3* gene, the literature contained no information regarding the correlation between the four SNPs linked to depression by our method and depression. However, several of them were evaluated for psychiatric and other behavioral disorders. Attention deficit hyperactivity disorder (ADHD) was associated with the rs13189021 variant [29,30]. SNP rs10052016 was associated with cognitive performance and alcohol dependence in the Irish Affected Sib Pair Study of Alcohol Dependence [31], whereas SNP rs10053602 was associated with cognitive performance in the Diabetes Heart Study, a familially enriched cohort with Type 2 Diabetes [32].

The *NR3C1* gene presented 26 variants significantly related to depression by the correlation method, but only two SNPs (rs41423247 and rs1866388) had been previously evaluated in relation to depression in the scientific literature. Two case–control studies on depression evaluated SNP rs41423247, with contradictory results. In contrast to the findings by Tan et al. [33] regarding perinatal depression, the homozygous genotype for the minor allele was significantly associated with the hippocampus shape and the integrity of the white matter of the para-hippocampal cingulate in a study of patients with major depressive disorder [34]. Regarding the rs1866388 SNP, Zobel et al. [35] analyzed seven *NR3C1* gene SNPs in a German population and discovered a statistical association between rs1866388 (GG genotype) and depression, corroborating our findings.

As a large number of *NR3C1* gene variants are associated with depression, we performed linkage analyses between these SNPs using the LDlink tool, resulting in the discovery of four independent alleles and two linkage blocks. The same method applied to the *SLC6A3* gene variants revealed four allelic variants, two with low linkage disequilibrium (rs13189021 and rs62331084) and two with complete linkage disequilibrium (rs13189021 and rs62331084). Importantly, many of the variants that showed a positive correlation in our study are not included in the sequencing platforms that are typically used for GWAS or are only included in a few of them. Consequently, the procedure herein proposed can provide more information about genetic links to diseases, even about variants that were not, or insufficiently, examined in GWAS.

Finally, a recent study by Couto-Silva et al. [36] can be used as an example of an approach in which epidemiological data and the frequencies of a given variant in different populations of humans can generate very interesting results. The authors were able to demonstrate an intense natural selection signal in a set of genes related to *Trypanosoma cruzi* infection by analyzing genomic data from 19 Amazonian native populations and cross-referencing variant frequency data with Chagas disease incidence data in this region. Such a result suggests that approaches employing methodologies based on similar precepts as the one herein described might be useful for identifying genetic factors associated with susceptibility to infectious diseases.

## 5. Conclusions

In conclusion, the findings of this study suggest that it may be possible to identify genetic markers associated with specific diseases if the allele and genotype frequencies are statistically analyzed and compared to epidemiological data about these diseases in various human populations. This method can be used as a supplement to evaluate the results of genome-wide association studies (GWAS). It can also be used to evaluate variants that are not included in GWAS platforms, providing us with new and useful information about targets, genes or variants that can be investigated as potential biomarkers in various circumstances. Compiling different studies of human genomes into a single database that groups individuals sequenced by country, allowing researchers to enter the phenotypic frequencies of the trait to be evaluated and returning the possible genetic variants correlated with the trait to be evaluated, would be of great interest. We emphasize the significance of linkage analysis for interpreting the output of the correlation method. Finally, it is important to highlight that this is a study performed using like data stored in several distinct databases, and therefore, our interpretations are dependent on how accurate these databanks are. Moreover, as a proposal mainly based in approaches in silico, our results should be taken with caution and mainly as suggestive of new targets for more conclusive experimental and population studies.

**Supplementary Materials:** The following supporting information can be downloaded at: https://www.mdpi.com/article/10.3390/biomedinformatics3020032/s1, Table S1—Pearson's correlation test with the GWAS variants; Table S2—Pearson's correlation test with the Endogenous genes-PPIA variants; Table S3—Pearson's correlation test with the *TPH2* variants; Table S4—Pearson's correlation test with the *NR3C1* variants; Table S5—Pearson's correlation test with the *SLC6A2* variants; and Table S6—Pearson's correlation test with the *SLC6A3* variants.

**Author Contributions:** Conceptualization, J.A.B.C. and A.D.C.; methodology, S.M.D.R. and A.D.C.; formal analysis, S.M.D.R., C.A.B. and A.D.C.; writing—original draft preparation, S.M.D.R.; writing—review and editing, C.A.B.; J.A.B.C. and A.D.C.; supervision, J.A.B.C. project administration, A.D.C. All authors have read and agreed to the published version of the manuscript.

**Funding:** This research received no external funding.

**Data Availability Statement:** All data is available under request.

**Conflicts of Interest:** The authors declare no competing interests.

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
