# Peer review of "Uncovering Disease-Related Polymorphisms through Correlations between SNP Frequencies, Population and Epidemiological Data"

_biomedinformatics, doi:10.3390/biomedinformatics3020032_

Round 1
Reviewer 1 Report
The manuscript by Reis SD et al demonstrated the correlations between SNP and epidemiological data on depression. Several suggestions are below:
1. The content is not sufficient. There should be more experiments and controls. There’s not enough background on how these 3 groups of genes are selected. How the correlation results support the conclusion.
2. The paper is difficult to read. English editing is recommended
Author Response
Firstly, we thank the Reviewers for the time devoted to the evaluation of our manuscript and for the valuable comments. The suggestions have significantly increased the quality of our article.
The Authors’ responses are highlighted in this file, and all modifications in the manuscript were highlighted in yellow.
#Reviewer 1
The content is not sufficient. There should be more experiments and controls. There’s not enough background on how these 3 groups of genes are selected. How the correlation results support the conclusion.
Answer: Although at a first view it could seems that experiments and controls were performed in few individuals, this was not the case. Actually, three groups of genes were evaluated: 82 variants related to depression identified through GWAS, 331 variants in six genes not related with depression [ACTB (21 SNPs), B2M (36 SNPs), EEF2 (68 SNPs), GAPDH (39 SNPs), PLEK2 (121 SNPs), PPIA (46 SNPs)] ; and variants in four genes already tested by genotyping in case-control studies involving depression [TPH2 (550 SNPs), NR3C1 (562 SNPs), SLC6A2 (302 SNPs), and SLCA3 (377 SNPs)). The frequencies of all these SNPs were evaluated, in correlation with epidemiological data concerning depression occurrence all around the world. All such numbers are stated along the manuscript. The results show that it is possible to suggest potential genetic markers associated with specific pathologies, using the correlation between allelic and genotypic frequencies and the epidemiological data of this disease in different populations. We hope that, considering these information, the reviewer could reconsider our study.
- The paper is difficult to read. English editing is recommended
Response: The manuscript was revised considering grammar and content presentation. Finally, we thank Reviewer 1 again for their comments on our manuscript.

Reviewer 2 Report
The paper deals with important topics in BioMedInformatics. The authors have presented an alternative approach to identify disease-causing variants focused on the simultaneous evaluation of genome variant data obtained from databases and epidemiological data of a pathology.
However, I have a number of suggestions:
1. I would suggest reinforcing comparison with images, not only table representation. To make it clearer for future readers.
2. Please optimize the table sizing in the paper to meet the guidelines.
3. I would suggest reinforcing Conclusion section with ideas for future development and implementation.
4. Please provide a link to an open-access repository on which you conducted your research(comparison), and add them into references.
5. Some of the references are outdated and unlinked. Please fix it by using 3-5 years old papers in high-impact journals.
Author Response
Firstly, we thank the Reviewers for the time devoted to the evaluation of our manuscript and for the valuable comments. The suggestions have significantly increased the quality of our article.
The Authors’ responses are highlighted in this file, and all modifications in the manuscript were highlighted in yellow.
#Reviewer 2
The paper deals with important topics in BioMedInformatics. The authors have presented an alternative approach to identify disease-causing variants focused on the simultaneous evaluation of genome variant data obtained from databases and epidemiological data of a pathology.
However, I have a number of suggestions:
- I would suggest reinforcing comparison with images, not only table representation. To make it clearer for future readers.
Response: A flowchart was inserted showing the methodology used in the article.
- Please optimize the table sizing in the paper to meet the guidelines.
Response: Changed as suggested.
- I would suggest reinforcing Conclusion section with ideas for future development and implementation.
Response: Changed as suggested.
- Please provide a link to an open-access repository on which you conducted your research(comparison), and add them into references.
Response: Changed as suggested.
- Some of the references are outdated and unlinked. Please fix it by using 3-5 years old papers in high-impact journals.
Response: We revised the references and included a recent manuscript from Couto-Silva et al (2023) as an example of approach where epidemiological data and the frequencies of a given variant in different human populations could drive quite interesting results.
Finally, we thank Reviewer 2 again for their comments on our manuscript.

Reviewer 3 Report
Uncovering disease-related polymorphisms through correlations between SNP frequencies, population and epidemiological data.
In this manuscript Samara Marques Reis et al. they propose an alternative method for identification of potential disease-causing variants. The authors focused on the simultaneous evaluation of genome variant data obtained from databases (mainly form the GWAS studies) and epidemiological data of a pathology. Unfortunately, GWAS cannot explain specific biological features of the disease, such as the physiological mechanisms beyond a given association. It seems that proposed approach may be another, additional and useful tool for identifying potential genetic risk variants in complex diseases.
Author Response
Firstly, we thank the Reviewers for the time devoted to the evaluation of our manuscript and for the valuable comments. The suggestions have significantly increased the quality of our article.
The Authors’ responses are highlighted in this file, and all modifications in the manuscript were highlighted in yellow.
#Reviewer 3
Uncovering disease-related polymorphisms through correlations between SNP frequencies, population and epidemiological data.
In this manuscript Samara Marques Reis et al. they propose an alternative method for identification of potential disease-causing variants. The authors focused on the simultaneous evaluation of genome variant data obtained from databases (mainly form the GWAS studies) and epidemiological data of a pathology. Unfortunately, GWAS cannot explain specific biological features of the disease, such as the physiological mechanisms beyond a given association. It seems that proposed approach may be another, additional and useful tool for identifying potential genetic risk variants in complex diseases.
Response: We thank Reviewer 3 again for comments on our manuscript.
